# LANGUAGE REWARD MODULATION FOR PRETRAINING REINFORCEMENT LEARNING

## ABSTRACT

Using learned reward functions (LRFs) as a means to solve sparse-reward reinforcement learning (RL) tasks has yielded some steady progress in task-complexity through the years. In this work, we question whether today's LRFs are best-suited as a direct replacement for task rewards. Instead, we propose leveraging the capabilities of LRFs as a pretraining signal for RL. Concretely, we propose **LA**nguage Reward **M**odulated **P**retraining (LAMP) which leverages the zero-shot capabilities of Vision-Language Models (VLMs) as a *pretraining* utility for RL as opposed to a downstream task reward. LAMP uses a frozen, pretrained VLM to scalably generate noisy, albeit shaped exploration rewards by computing the contrastive alignment between a highly diverse collection of language instructions and the image observations of an agent in its pretraining environment. LAMP optimizes these rewards in conjunction with standard novelty-seeking exploration rewards with reinforcement learning to acquire a language-conditioned, pretrained policy. Our VLM pretraining approach, which is a departure from previous attempts to use LRFs, can warmstart sample-efficient learning on robot manipulation tasks in RLBench.

## 1 INTRODUCTION

A longstanding challenge in reinforcement learning is specifying reward functions. Extensive domain knowledge and ad-hoc tuning are often required in order to manually design rewards that "just work." However, such rewards can be highly uninterpretable and riddled with cryptic mathematical expressions and constants. Furthermore, hand-crafted reward functions are often over-engineered to the domain in which they were designed, failing to generalize to new agents and new environments (Shah et al., 2022; Langosco et al., 2022). As a result, a long history of foundational work in Inverse Reinforcement Learning (IRL) (Abbeel & Ng, 2004; Ng & Russell, 2000; Kim et al., 2003; Peng et al., 2021; Escontrela et al., 2022) has produced an abundance of methods for learning rewards from demonstration data assumed to optimal under the desired reward function. However, learned reward functions are also notorious for noise and reward misspecification errors (Amodei et al., 2016; Amodei & Clark, 2016) which can render them highly unreliable for learning robust policies with reinforcement learning. This is especially problematic in more complex task domains such as robotic manipulation, particularly when in-domain data for learning the reward function is limited.

While acknowledging that learned reward functions are subject to potential errors, we hypothesize that they may be effectively employed to facilitate exploration during the lower-stakes pretraining stage of training. During pretraining, we generally desire a scalable means of generating diverse behaviors to warmstart a broad range of possible downstream tasks. LRFs are an appealing means for supervising these behaviors since they do not rely on human-design and carry the potential to scale with dataset diversity. Despite this potential, obtaining LRFs that generalize to new domains is non-trivial (Shah et al., 2022). Notably, however, large-pretrained models have shown impressive zero-shot generalization capabilities that enable them to be readily applied in unseen domains. Indeed, large pretrained VLMs have shown recent successes in reward specification for task learning by computing alignment scores between the image observations of an agent and a language input describing the desired task (Cui et al., 2022; Mahmoudieh et al., 2022). While these methods adopt similar reward misspecification shortcomings to other LRFs, they come with the novel and relatively under-explored property of being a scalable means of generating many *different* rewards by simply prompting with different language instructions. This property is particularly compatible

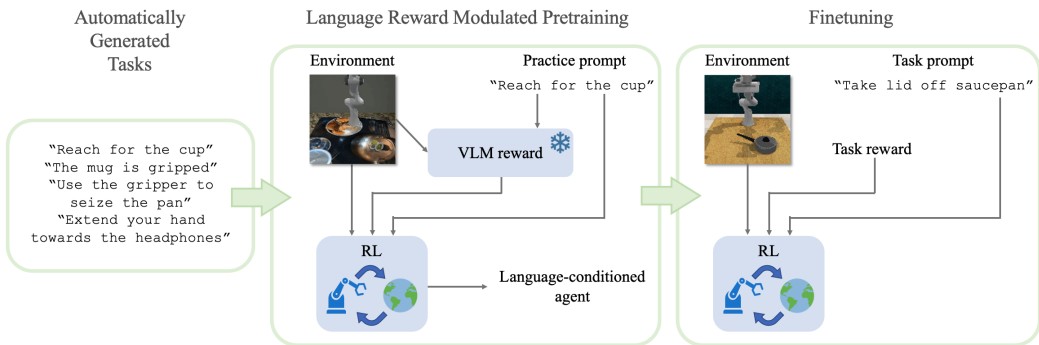

Figure 1: **LAMP Framework**. Given a diverse set of tasks generated by hand or by a LLM, we extract VLM rewards for language-conditioned RL pretraining. At finetuning time, we condition the agent on the new task language embedding and train on the task reward.

with the assumptions of RL pretraining where we desire to learn general-purpose behaviors with high coverage of environment affordances, require minimal human supervision, and do not require pretrained behaviors to transfer zero-shot to fully solve downstream tasks. Instead of relying on noisy VLM LRFs to train task-specific experts, can we instead use them as a tool for pretraining a general-purpose agent?

In this work, we investigate how to use the flexibility of VLMs as a means of scalable reward generation to *pretrain* an RL agent for accelerated downstream learning. We propose **LA**nguage Reward **M**odulated **P**retraning (LAMP), a method for pretraining diverse policies by optimizing VLM parameterized rewards. Our core insight is that instead of scripting rewards or relying solely on general unsupervised objectives to produce them, we can instead query a VLM with highly diverse language prompts and the visual observations of the agent to generate diverse, shaped pretraining rewards. We augment these rewards with intrinsic rewards from Plan2Explore (Sekar et al., 2020), a novelty-seeking unsupervised RL algorithm, resulting in an objective that biases exploration towards semantically meaningful visual affordances. A simple language-conditioned, multitask reinforcement learning algorithm optimizes these rewards resulting in a language-conditioned policy that can be finetuned for accelerated downstream task learning. We demonstrate that by pretraining with VLM rewards in a visually complex environment with diverse objects, we can learn a general-purpose policy that more effectively reduces the sample-complexity of downstream RL. We train LAMP in a pretraining environment with realistic visual textures and challenging randomization and evaluate downstream performance on RLBench tasks. We also analyze the influence of various prompting techniques and frozen VLMs on the performance of the pretrained policy.

## 2 RELATED WORK

**Pretraining for RL** Following on the successes of pretraining in vision and language, a number of approaches have grown in interest for pretraining generalist RL agents in order to reduce sample-complexity on unseen tasks. Classical works in option learning (Sutton et al., 1999; Stolle & Precup, 2002) and more recent approaches in skill discovery such as (Sharma et al., 2019; Gregor et al., 2016; Park et al., 2023; Eysenbach et al., 2018) look to pretrain skill policies that can be finetuned to downstream tasks. Exploration RL algorithms such as (Burda et al., 2018; Sekar et al., 2020; Pathak et al., 2017; Liu & Abbeel, 2021) use unsupervised objectives to encourage policies to learn exploratory behaviors. Works such as (Xiao et al., 2022; Nair et al., 2022; Radosavovic et al., 2023) leverage pretrained vision encoders to accelerate RL from pixels. LAMP combines a large-pretrained VLM with exploration-based RL to guide exploration towards meaningful behaviors.

**Inverse RL from human video** Inverse reinforcement learning (IRL) (Arora & Doshi, 2021; Ziebart et al., 2008; Abbeel & Ng, 2004; Ng & Russell, 2000) proposes a number of approaches to address the challenges associated with learning reward functions from demonstrations. A number of more recent works focus on inverse RL from video datasets of human interaction, (Chen et al., 2021; Zakka et al., 2021; Sermanet et al., 2016; 2017) which are often more readily available than

in-domain demonstrations. These methods rely on perceptual metrics such as goal-image similarity and trajectory similarity to formulate rewards but require task-specific paired data. Other methods such as (Ma et al., 2023) make weaker assumptions on the task-specificity of the human video dataset and thus can leverage "in-the-wild" data and exhibit stronger domain generalization. LAMP similarly exploits "in-the-wild" video data via a frozen, pretrained VLM but focuses on leveraging language to flexibly modulate the VLM and generate diverse rewards.

**VLMs as task rewards** A number of works propose methods for extracting shaped task reward signals from large-scale pretrained LLMs or VLMs (Du et al., 2023b; Cui et al., 2022; Mahmoudieh et al., 2022). Others such as (Du et al., 2023a) leverage pretrained VLMs as general-purpose success detectors which can provide sparse task rewards. VLM-based rewards can also be learned on bespoke datasets or in-domain data (Fan et al., 2022; Shao et al., 2020) to be used directly for downstream task learning. Instead of relying on VLMs to train task-specific experts, LAMP uses a VLM to control scalable RL pretraining and employs scripted task rewards to demonstrate reliable downstream finetuning.

## 3 BACKGROUND

**Reinforcement learning** We consider the reinforcement learning (RL) framework where an agent receives an observation $o_t$ from an environment and chooses an action $a_t$ with a policy $\pi$ to interact with the environment. Then the agent receives an extrinsic reward $r_t^{\text{e}}$ and a next observation $o_{t+1}$ from the environment. The goal of RL is to train the policy to maximize the expected return defined as a cumulative sum of the reward with a discount factor $\gamma$, *i.e.*, $\mathcal{R}_t = \mathbb{E}[\sum_{k=0}^{\infty} \gamma^k r^{\text{e}}(o_{t+k}, a_{t+k})]$.

**Reinforcement learning with vision-language reward** In sparse reward tasks, the extrinsic reward $r^{\text{e}}$ becomes non-zero only when the task successfully terminates, making it difficult to learn policies that complete the target tasks. To address this, recent approaches have proposed to use the vision-language alignment score from a large-scale vision-language model (VLM) as a reward (Fan et al., 2022; Cui et al., 2022). Formally, let $\mathbf{x} := \{x_1, ..., x_M\}$ be a text that describes the task consisting of $M$ tokens, $F_\phi$ be a visual feature encoder, and $L_\alpha$ be a language encoder. Given a sequence of transitions $\{o_i, a_i, r_i^{\text{e}}, o_{i+1}\}_{i=1}^N$, the key idea is to use the distance between visual representations $F_\phi(o_i)$ and text representations $L_\alpha(\mathbf{x})$ as an intrinsic reward, which is defined as $r_i^{\text{int}} = D\left(F_\phi(o_i), L_\alpha(\mathbf{x})\right)$, where $D$ can be an arbitrary distance metric such as cosine similarity or L2 distance. This intuitively can be seen as representing the extent to which the current observation is close to achieving the task specified by the text.

**R3M** The vision encoders of video-language models have been successfully employed as semantic feature extractors that enable downstream learning on a variety of domains including standard prediction and classification tasks as well as, more recently, decision making and control (Xu et al., 2021). Notably, R3M, has lead to improvements in the data-efficiency of imitation learning in real-world robotics domains (Nair et al., 2022). R3M extracts semantic representations from the large-scale Ego4D dataset of language annotated egocentric human videos (Grauman et al., 2022). The language input is processed by $L_\alpha$, a pretrained DistilBERT transformer architecture (Sanh et al., 2019) that aggregates the embeddings of each word in the instruction and the images are encoded with R3M's pretrained ResNet-18 $F_\phi$. A video-language alignment loss encourages the image representations $F_\phi(\cdot)$ to capture visual semantics by extracting image features that aid in predicting the associated language annotations, which are embedded by $L_\alpha(\mathbf{x})$. In particular, R3M trains $G_\theta(F_\phi(o_1), F_\phi(o_i), L_\alpha(\mathbf{x}))$ to score whether the language $\mathbf{x}$ explains the behavior from image $o_1$ to image $o_i$. The score function is trained simultaneously to the representations described above with a contrastive loss that encourages scores to increase over the course of the video and scores to be higher for correct pairings of video and language than incorrect ones.

## 4 METHOD

We present **LA**nguage **R**eward **M**odulated **P**retraining (LAMP), a simple yet effective framework for pretraining reinforcement learning with intrinsic rewards modulated with language instructions. LAMP consists of two stages: (i) a task-agnostic RL pretraining phase that trains policies to maximize

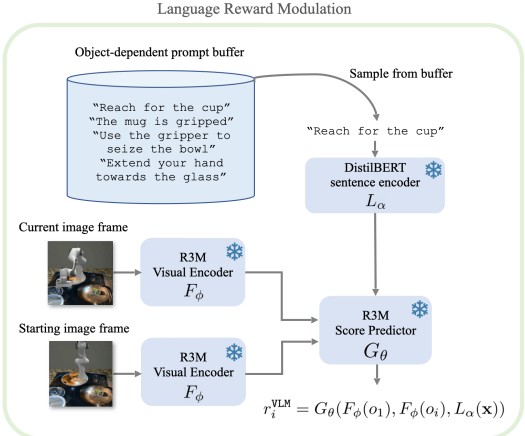

Figure 2: **LAMP Method**. We use R3M (Nair et al., 2022) for our VLM-based rewards. We query the R3M score predictor for pixel and language alignment, which is pretrained on the Ego4D dataset (Grauman et al., 2022). The reward model is frozen.

the VLM-based rewards and (ii) a downstream task finetuning phase that adapts pre-trained policies to solve the target tasks by maximizing the task reward. In this section, we first describe how we define our intrinsic reward (see Section 4.1), how we pretrain policies (see Section 4.2), and how we adapt the policies to downstream tasks (see Section 4.3). We provide the overview and pseudocode of LAMP in Figure 2 and Algorithm 1, respectively.

## 4.1 LANGUAGE REWARD MODULATION

**R3M score as a reward** To extract pretraining reward signals for RL, we propose to use the R3M score as a source of rewards. Our motivation is that the R3M score is well-suited for providing shaped rewards because its representations are explicitly trained to understand temporal information within videos (see Section 3 for details) in contrast to other VLMs without such components (Radford et al., 2021; Xu et al., 2021; Wang et al., 2022). Specifically, we define our VLM reward using the R3M score as below:

$$r_i^{\texttt{VLM}} = G_\theta(F_\phi(o_1), F_\phi(o_i), L_\alpha(\mathbf{x})) \tag{1}$$

where $G_\theta$ denotes the score predictor in R3M. Intuitively, this reward measures how $o_i$ is making a progress from $o_1$ towards achieving the tasks specified by natural language instruction $\mathbf{x}$. We find that our reward is indeed better aligned with the progress within expert demonstrations in our considered setups compared to other VLMs (see Figure 3 for supporting experiments).

**Rewards with diverse language prompts** To fully exploit the language understanding of VLMs, we propose to query them with a diversity of texts describing a diversity of objectives, as opposed to computing the reward by repeatedly using a single instruction. Specifically, we obtain diverse, semantic rewards modulated by language, generating diverse sets of language instructions for each task and use them for prompting the model. Given an instruction template, we query ChatGPT[1] for a diverse set of language instructions with a focus on two categories: imperative instructions and statements of completion (*e.g.* move the mug vs. the mug is moved). Given that large-scale video datasets are predominantly human-centric, we obtain prompts that are human-centric, robot centric, as well as ambiguous (*e.g.* the robot arm moved the mug vs. use your hand to move the mug vs. reach toward the mug and move it). Moreover, we augment the instructions by querying for synonym nouns. By inputting diverse language instructions from the dataset along with the agent's image observations, we effectively modulate the frozen, pretrained R3M reward and produce diverse semantic rewards that are grounded in the visual environment of the agent.

---

[1] https://chat.openai.com

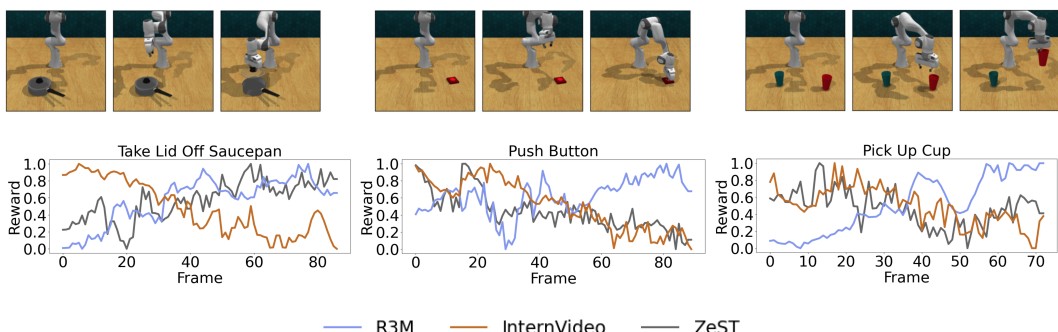

Figure 3: Video-Language alignment scores from R3M (Nair et al., 2022), InternVideo (Wang et al., 2022), and ZeST (Cui et al., 2022) on RLBench downstream tasks plotted over an expert episode with 3 snapshots visualized. Rewards are highly noisy and do not increase smoothly throughout the episode. Optimizing this signal with RL is unlikely to lead to stable solutions, and thus we instead use rewards as an exploration signal during pretraining.

## 4.2 LANGUAGE-CONDITIONED PRETRAINING

While several recent works have shown that rewards from VLMs can be used for training RL agents (Fan et al., 2022; Cui et al., 2022), it is still questionable whether these rewards can serve as a sufficiently reliable signal for inducing the intended behavior. In this work, we instead propose to leverage such rewards from VLMs as a pretraining signal for RL policies, utilizing the knowledge captured within large VLMs for scripting diverse and useful exploration rewards.

**Pretraining environment**   To learn diverse behaviors that can be transferred to various downstream tasks, we design a set of tasks with realistic visuals and diverse objects. Specifically, we build a custom environment based on the RLBench simulation toolkit (James et al., 2020). In order to simulate a realistic visual scene, we download images from the Ego4D dataset (Grauman et al., 2022) and overlay them as textures on the tabletop and background of the environment (see Figure 2). To produce diverse objects and affordances, we import ShapeNet (Chang et al., 2015) object meshes into the environment. Both the visual textures and the objects are randomized every episode of training.

**Objective**   Because the VLM reward can be seen as measuring the extent to which the agent is closer to solving the task (see Section 4.1), it can be readily be combined with novelty-seeking unsupervised RL methods that optimize both extrinsic and intrinsic rewards. Therefore, to incentivize exploration, we combine the VLM reward with the novelty score from a separate exploration technique. Specifically, we consider Plan2Explore (Sekar et al., 2020) that utilizes the disagreement between future latent state predictions as a novelty score. Let this novelty-based score be $r_i^{\text{P2E}}$. We then train our pretraining agent to maximize the following weighted sum of rewards:

$$r_i^{\text{LAMP}} = \alpha \cdot r_i^{\text{P2E}} + (1 - \alpha) \cdot r_i^{\text{VLM}} \tag{2}$$

where $\alpha$ is a hyperparameter that balances the two rewards. By combining this novelty-based reward with the VLM reward, we encourage the agent to efficiently explore its environment but with an additional bias towards interacting with the semantically meaningful affordances. We found that an $\alpha$ value of 0.9 works quite well across the tasks evaluated.

**Pretraining pipeline**   During task-agnostic pretraining, the agent is deployed in a language-conditioned MDP where there are no environment task rewards $r_i^{\text{e}}$. For the underlying RL algorithm, we use Masked World Models (MWM) (Seo et al., 2022), an off-policy, model-based method with architectural inductive biases suitable for fine-grained robot manipulation. Every episode, our method randomly samples some language prompt $\mathbf{x}$ from the generated dataset as specified in Section 4.1. Then we condition the MDP and agent on the respective embedding $L_\alpha(\mathbf{x})$ such that each rolled out transition in the episode can be expressed as $(o_i, a_i, o_{i+1}, L_\alpha(\mathbf{x}))$. After each episode is collected, we compute the rewards for each transition by embedding the observations with the R3M visual encoder

---

**Algorithm 1** Language Reward Modulated Pretraining (LAMP)

---

1: Initialize Masked World Models (MWM) parameters
2: Load pretrained DistilBERT $L_\alpha$
3: Load pretrained R3M visual encoder $F_\phi$
4: Load pretrained R3M score predictor $G_\theta$
5: Initialize replay buffer $\mathcal{B} \leftarrow 0$
6: Prefill language prompt buffer $\mathcal{B}^l$
7: Prefill synonym buffer $\mathcal{B}^s$
8: **for** each episode **do**
9:     Randomize scene textures by sampling among Ego4D and original RLBench textures
10:     Sample ShapeNet Objects to place in scene
11:     Sample language prompt $\mathbf{x}$ from $\mathcal{B}^l$ (e.g., `Pick up the [NOUN]`)
12:     Replace `[NOUN]` in $\mathbf{x}$ by sampling a synonym from $\mathcal{B}^s$ for a random object in the scene
13:     Process the prompt via DistilBERT to obtain language embeddings $L_\alpha(\mathbf{x})$
14:     Collect episode transitions with $\pi(a|(s, L_\alpha(\mathbf{x}))$
15:     Assign LAMP rewards to all time steps (in parallel) by embedding frames with $F_\phi$ and querying the R3M score predictor $G_\theta$
16:     Add all episode transitions to $\mathcal{B}$
17:     Update MWM and Plan2Explore parameters as in (Seo et al., 2022; Sekar et al., 2020) by sampling transitions from $\mathcal{B}$ and augmenting LAMP rewards with novelty bonus to train agent
18: **end for**

---

$F_\phi$ and then applying the R3M score predictor $G_\theta$ - afterwards adding the data to the replay buffer. We then sample batches of transitions from the buffer, augment the reward with the Plan2Explore intrinsic reward, and make reinforcement learning updates to a language-conditioned policy, critic, and world model. By the conclusion of pretraining, we obtain a language-conditioned policy capable of bootstrapping diverse behaviors specified by the language $\mathbf{x}$.

### 4.3 DOWNSTREAM TASK ADAPTATION

In order to evaluate the quality of the pretrained skills, we evaluate on downstream reinforcement learning tasks with scripted task rewards $r_i^e$. Since we have learned a language-conditioned policy, we simply select a language instruction $\mathbf{x}_{ft}$ roughly corresponding to the downstream task semantics in order to condition the pretrained agent. We remark that an additional advantage of LAMP is its use of language as a task-specifier, which enables this simplicity of zero-shot selection of a policy to finetune (Adeniji et al., 2022). We fix this language instruction selection for the entirety of task learning and finetune all RL agent model components except the critic, which we linear probe for training stability.

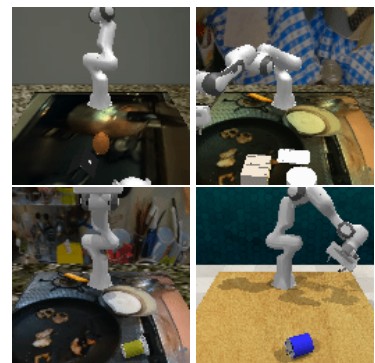

Figure 4: We pretrain on domain-randomized environments based on Ego4D textures, occasionally sampling the default, non-randomized RLBench environment.

## 5 EXPERIMENTS

### 5.1 SETUP

**Environment details**    As previously mentioned in Section 4.2, we consider domain-randomized environments for pre-training (see Figure 4 for examples).

Specifically, our pretraining environments consist of 96 domain-randomized environments with different Ego4D textures overlayed over the table, walls, and floor. We also sample the environments having default RLBench environment textures with probability of 0.2. We do not include any of the downstream evaluation tasks that the agent finetunes on in the pretraining setup. The pretraining tasks are exclusively comprised of ShapeNet objects. All of the downstream tasks are unseen tasks from the original RLBench task suite.

Figure 5: Finetuning performance on visual robotic manipulation tasks in RLBench. We include results with additional unsupervised RL baselines in the supplementary material. The solid line and shaded region represent mean and standard deviation across 3 seeds.

For finetuning, we implement a shaped reward function based on the ground truth simulator state and train the agent to optimize this signal instead of the pretraining reward. We use the exact scenes released in RLBench in order to encourage reproducibility, notably keeping the default background and table textures fixed throughout the course of training. We use a 4-dimensional continuous action space where the first three dimensions denote end-effector positional displacements and the last controls the gripper action. We fix the end-effector rotation and thus select tasks that can be solved without changing the orientation of the end-effector.

**Baselines** As a baseline, we first consider a randomly initialized MWM agent trained from scratch to evaluate the benefit of pretraining. In order to evaluate the benefit of pretraining with LAMP that modulates the reward with language, we also consider Plan2Explore (Sekar et al., 2020) as a baseline, which is a novelty-seeking method that explores based on model-uncertainty. Our method, LAMP, is the combination of the VLM reward and the Plan2Explore reward with a fixed $\alpha$ value of 0.9 across tasks as described in equation 2.

## 5.2 RESULTS

Across tasks in Figure 5, we find that training a randomly initialized agent from scratch on a new task exhibits high-sample complexity in order to learn a performant policy. Across most RLBench tasks, Plan2Explore, which employs purely unsupervised exploration, exceeds the performance of training from scratch. LAMP outperforms or is competitive with Plan2Explore and consistently outperforms training from scratch. We hypothesize that this is because by pretraining with a VLM reward, LAMP biases the agent to explore efficiently in ways that are semantic meaningful and avoids spending time investigating spurious novelty. It is also possible that by optimizing more diverse rewards during pretraining, the agent learns representations that allow it to quickly adapt to the unseen task reward during finetuning. We further note that the VLM rewards in isolation can lead to effective pretraining as we demonstrate in Figure 8. Solely optimizing the VLM rewards (LAMP w/o Plan2Explore) is able to achieve strong performance, however, we find that the combination of VLM and Plan2Explore rewards yields the best overall performance.

## 6 ABLATIONS

### 6.1 LANGUAGE PROMPTING

A particular convenience of using VLMs trained on internet-scale data is the diversity of language we can query for plausibly infinite reward functions. We ablate different prompt styles used during the pretraining phase. The 6 language prompting styles are as such:

1. **Prompt Style 1**: `Pick up the [NOUN].`
2. **Prompt Style 2**: `[RELEVANT VERB] and [SYNONYM NOUN].`
3. **Prompt Style 3**: `[RELEVANT VERB] and [RANDOM NOUN].`
4. **Prompt Style 4**: `[IRRELEVANT VERB] and [SYNONYM NOUN].`
5. **Prompt Style 5**: `[IRRELEVANT VERB] and [RANDOM NOUN].`
6. **Prompt Style 6**: `[Snippets from Shakespeare].`

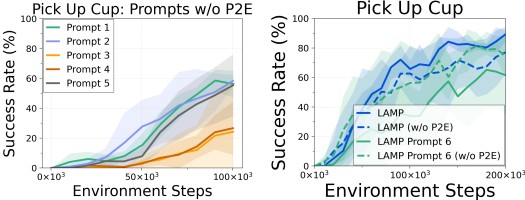

Figure 6: Finetuning performance on RLBench task. (Left) Effect of pretraining with rewards from different language prompting styles. Language prompts focus on action-based tasks. (Right) Effect of pretraining on action-based prompts (Lang 2) and random prompts (Lang 6).

For Prompt Styles 1-5, we compare the effect of using relevant and irrelevant nouns and verbs, though all remain action-based and task-relevant. For Prompt Style 6, we select snippets of Shakespeare text to see the effect of pretraining on rewards generated from completely out of distribution and implausible tasks.

The prompting styles provide varying types of language diversity during pretraining. We evaluate how important it is that the prompts be aligned with possible tasks in the environment– might an instruction like "sweep the floor" still encourage possibly more interesting behaviors even when the agent is only presented with mugs? In addition, providing a multitude of prompts may mitigate the adverse effects of overfitting by exploiting the imperfections of the VLM rewards for a particular prompt.

In Figure 6 (Left), we compare the finetuning results of Prompt Styles 1-5, which are action-based prompts. We evaluate on the task "Pick Up Cup" because the task name is simple and aligned to prompts used during pretraining. We find that for this task, Prompt Style 2, which pretrains on semantically similar but diverse wordings of prompts, is most successful. In addition, Prompt Style 1, which pretrains on very simple instructions that are similar to the pretraining task, finetunes efficiently, as well. For our main experiments, we choose Prompt 2 based on both its strong finetuning performance as well as increased diversity compared to Style 1.

In Figure 6 (Right), we also compare the performance of our best action-based prompt, Prompt 2, with a non-action-based prompt, Prompt 6. In this figure, we also investigate the effect of the auxiliary exploration objectives. While LAMP Prompt 6 (w/o Plan2Explore) and LAMP Prompt 2 (w/o Plan2Explore) perform similarly, we notice that adding in the exploration objectives dramatically decreases the finetuning effectiveness of LAMP Prompt 6. We hypothesize that both exploration coverage and exploration efficiency during pretraining play an important role. By separately increasing exploration coverage through Plan2Explore, the quality of the VLM rewards becomes more important for focusing the auxiliary exploration objective on useful exploratory behaviors. Thus, LAMP Prompt 2, which incorporates Plan2Explore, is trained on higher-quality, more relevant rewards, and can explore more efficiently during pretraining, and therefore enables more effective finetuning.

Overall, the differences in pretraining with different action-based prompting styles is not extreme, suggesting that LAMP is robust to different prompting strategies, and providing diverse instructions can be an effective way of pretraining an agent.

## 6.2 REWARD WEIGHTING ABLATION

We provide an ablation of the $\alpha$ reward weighting term for the joint pretraining reward introduced in Equation 2. As shown in Figure 7, we find that the method is somewhat robust to the choice of $\alpha$, however, larger values in general work better. We also isolate the relative contributions of the Plan2Explore and LAMP rewards by setting the alpha value to 0 and 1. We find that, while either option performs somewhat similarly, the synergy of the two rewards achieves the highest performance.

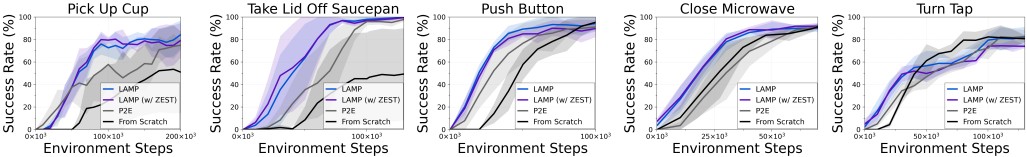

Figure 7: We find that LAMP is robust to the reward weighting hyperparameter alpha. Notably, with purely VLM rewards (alpha = 0), LAMP obtains competitive performance across all tasks evaluated. However, the synergy of the two rewards achieves the strongest performs.

## 6.3 VLM MODEL

We compare our method across different vision-language reward models. CLIP, which is trained on static image-caption pairs, can serve as a reward model by extracting the cosine similarity between text feature and image feature changes as presented as ZeST in (Cui et al., 2022). Following ZeST, in order to express feature displacements, we assign context features in image and language space as $s_0$, the initial image in an episode, and $\mathbf{x_0}$, a language prompt that inverts the action (open microwave inverts close microwave) described in the desired instruction $\mathbf{x}$. In particular, we use the following reward parameterization,

$$r_i = (F_\phi(s_i) - F_\phi(s_0)) \cdot (L_\alpha(\mathbf{x}) - L_\alpha(\mathbf{x_0})). \tag{3}$$

where $F_\phi$ is the CLIP pretrained image encoder, $L_\alpha$ is the CLIP pretrained language encoder, and the task reward $r_i$ is defined as the dot product of the visual and language delta embeddings.

We show finetuning results on the RLBench downstream tasks in Figure 8. While R3M seems to lead to consistently good performance, we observe that LAMP pretraining with CLIP rewards can also perform well.

Figure 8: Finetuning performance on visual robotic manipulation tasks in RLBench. The solid line and shaded region represent mean and standard deviation across 3 seeds. We observe that CLIP rewards can also work well with LAMP components.

## 7 DISCUSSION

In this work, we present LAMP, an algorithm for leveraging frozen VLMs to pretrain reinforcement learning agents. We observe a number of limitations of using LRFs for task reward specifcation in RL, and propose a new setting and methodology that is more accommodating of these limitations. We demonstrate that by leveraging the flexibility and zero-shot generalization capabilities of VLMs, we can easily produce diverse rewards during pretraining that encourage the learning of semantically grounded exploratory behaviors. Furthermore, we show that VLM parameterized rewards exhibit strong synergies with novelty-seeking RL pretraning methods and can lead to strong performance in combination with Plan2Explore. We evaluate LAMP on a number of challenging downstream tabletop manipulation tasks from RLBench as well as show evidence that our method is not limited to a particular VLM architecture. Overall, we see LAMP as an indication of the promise of leveraging large pretrained VLMs for pretraining behaviors in challenging environments. A limitation of LAMP is its reliance on performing inference of the VLM model many times throughout the course of pretraining in order to generate rewards. Inference speed for larger, more powerful VLMs may be slow, bottlenecking the speed of the pretraning process. Another limitation is that LAMP does not address the long-horizon sequencing setting where we might be interested in finetuning agents conditioned on many language prompts. We leave this as in intriguing direction for future work.

