# OpenReview forum: "Language Reward Modulation for Pretraining Reinforcement Learning"
_ICLR.cc/2024/Conference — Submitted to ICLR 2024_

### Official Review · Reviewer_GnSb · 2023-10-16

**Soundness:** 2 fair
**Presentation:** 2 fair
**Contribution:** 2 fair
**Rating:** 6
**Confidence:** 4

**Summary:**

Specifying the reward function is a longstanding yet fundamental problem in RL. While the learned reward functions (LRFs) can have errors, they may help exploration during the lower-stakes pretraining stage of training, where diverse, rather than task-specific, behaviors are desired to be learned. By the strong zero-shot ability of vision-language models (VLMs), one may utilize pre-trained VLMs to specify the different reward functions in different tasks. This paper investigates how to use the flexibility of VLMs to scalably generate rewards to pretrain an RL agent, so as to accelerate downstream learning when a specific task is given. The authors propose LAnguage Reward Modulated Pretraning (LAMP), a method for pretraining diverse policies by optimizing VLM parameterized rewards. In particular, they query a VLM with highly diverse language prompts and the visual observations of the agent to generate diverse pretraining rewards, which is further augmented with the intrinsic reward from Plan2Explore. Experimentally, the authors demonstrate that a general-purpose policy can be learned by pretraining the agent with VLM reward. Such a policy can speed up the downstream RL tasks.

**Strengths:**

1. The overall direction of language-conditional control and leveraging VLMs for (pre)training RL agent is promising and interesting.
2. The technique is generally novel of prompting VLMs with diverse synonyms language prompts to generate the (pretraining) rewards for an array of tasks.
3. It is interesting to see the ablation study on different prompt styles used during the pretraining phase.

**Weaknesses:**

1. It is not clear to me what is the main contribution of the proposed method. Many, if not most, of the components in the proposed method come from prior works, such as R3M, Plan2Explore, vision-language-embedding matching and so on. Meanwhile, the idea of conditioning the reward function on instructions/goals is also not novel, see the following Point 2.
2. It seems missing the connection of the proposed method with  goal-conditioned RL and and distinction from them. For example, how is prompting VLM with different language instructions different from inputting different goal states to the goal-conditioned reward function?
3. The proposed pre-training stage seems computationally expensive, requiring rollouts, language and visual embeddings, calling the R3M score predictors, and so on, before updating the RL components.
4. Considering the relatively wide error bars in Fig. 5, the performance gain of the proposed method LAMP may not be clear when comparing with the baseline P2E.
5. Figure 7 may not convincingly support the claim that the proposed method "is not inherently reliant on a particular VLM", since the authors only show the results of ZeST besides the main choice R3M.

**Questions:**

1. How does the performance of the proposed method compare with prior works that learns options/skills as pretraining? And what about compared with goal-conditioned RL methods, which may be used for pretraining similar to the proposed method?
2. Why is the pretraining stage lower-stakes? Do you have any reference or justification?
3. Is there any intuition why the $r^{VLM}$ is combined with the intrinsic rewards from Plan2Explore,  which is itself a novelty-seeking score? Is there any intuition why in Fig. 7 the performance degrades quite significantly when using $r^{VLM}$ alone to pre-train the agent?
4. The $R_t$ just below the title of Section 3 seems missing an expectation symbol?
5. Are there any requirements on the downstream tasks so that the pretraining pipeline in Section 4.2 remains helpful?
6. How sensitive is the proposed method to the $\alpha$ parameter in Eq. 2?
7. In Figure 6 (right), why does LAMP Prompt 6 (w/o P2E) perform so well, close to LAMP under Prompt Style 2 (even with P2E)? This is a bit counter-intuitive to me since Prompt 6 is "Snippets from Shakespeare".
8. Is there an ablation study on the novelty-seeking RL pretraning methods. For example, what if we change Plan2Explore to other methods?

---

> ### Author Response · Authors · 2023-11-13
>
> **Primary Contribution**
>
> The primary contribution of this work is to propose a possible route towards pretraining a single generalist RL agent with limited human supervision that can efficiently adapt to many new tasks. Toward this goal we select a variety of downstream finetuning tasks for evaluation and do not include any of these tasks in our pretraining setup nor do we enforce any very strict requirements on their connection to the pretraining tasks. R3M has been used as a visual encoder to accelerate behavior learning by initializing with generalizable representations, however, to our knowledge has not been used as a reward function. Furthermore, instead of using a VLM to specify a task reward for a single language instruction or image goal as is done in [2][3], we demonstrate experimentally that by using a diverse set of language prompts with a frozen VLM, LAMP can be pretrained to do a wide array of tasks even more effectively than with the most widely-used exploration algorithms (Plan2Explore, RND). Our pretraining design is a divergence from existing approaches both in our inclusion of zero-shot VLM rewards from representation learning models and the way in which we generate them with diverse language prompting styles with loose relevance to the affordances in the pretraining scene. We also demonstrate a synergy between these pretrained components and fully unsupervised methods which to our knowledge has not been experimentally demonstrated. Lastly, we conduct a systematic evaluation of which language promoting techniques are best suited for the pretraining setting, which we anticipate to be of interest to the community as VLMs grow in popularity.
>
> **Connection to Goal-Conditioned RL**: Our method is related to goal-conditioned RL in the sense that we learn a language-conditioned agent and language instructions can be considered semantic goals. However, the conventional goal-conditioned RL setup does not address the desiderata of scalable behavior pretraining. In general, this setting requires human supervision to select interesting goals for the agent to reach and actually achieving those goals often remains an exploration challenge. A contribution of LAMP is in identifying a compatibility of the relative ease of enumerating exploratory language goals compared to image goals and the constraints on human involvement for open-ended pretraining.
>
> **Pretraining Desiderata and Computational Cost**
>
> There have been considerable efforts to leverage VLMs such as [4] as a task reward. In fact, works such as [5] select datasets and architectures towards producing general-purpose reward functions. However, we argue in this work that due to issues of reward exploitation and noisy models that these LRF’s are ill-suited for directly learning downstream tasks. We instead argue that the pretraining phase is a more favorable context, which we refer to as “lower-stakes”, as we are reasonably less concerned with the sample-inefficiency of a “one-time cost” that requires limited human supervision than for a recurring cost we incur each time we desire to quickly learn a new skill. This is corroborated by [7]. Moreover, the differences in desiderata between pretraining and finetuning make the admittedly somewhat computationally intensive nature of LAMP more palatable. The paradigm of computationally expensive, large-scale pretraining in service of multi-purpose few-shot or task-specific finetuning characterizes much of modern machine learning [6].
>
> **Desired Results in Appendix**
>
> We were similarly interested to understand the impact of the reward weighting term. We had already provided these results in the Appendix. Please refer to Figure 11 in Section D.2 of the Supplementary Material. We find that LAMP works best with our choice of alpha but still exhibits very strong performance for an alpha value of 0 corresponding to training only with VLM rewards and no Plan2Explore bonus. We think this is quite an encouraging result and hope this addresses the reviewers concerns.
> We were additionally interested in how another novelty-seeking algorithm would perform. We direct the reviewer to an additional widely-used baseline, Random Network Distillation (RND) [1] in the Appendix (Figure 13, Section D.3.1). LAMP is able to outperform this baseline on all of the tasks evaluated.

---

> ### Author Response · Authors · 2023-11-13
>
> **Baselines**
>
> Knowledge-Based Exploration algorithms are indeed not the only class of pretraining methods. However, we direct the reader to [7]. URLB conducts a systematic review of the various classes of unsupervised pretraining methods and finds that across every task evaluated, the Knowledge-Based methods (Plan2Explore, RND, and Disagreement) outperform the Data-Based methods (APT, ProtoRL) and Competence-Based (Skill Learning) methods (APS, DIAYN, SMM). This holds across all tasks from state but is even more pronounced for the pixel-based tasks where for the robotics tasks, evaluated the Knowledge-Based methods achieve ~4 times the normalized return of the Data-Based methods and ~15 times the normalized return of the Competence-Based methods (See Figure 3 and Figure 4 of [7]). This evidence guided our evaluation and selection of baselines.
>
> [1] Exploration by Random Network Distillation (Burda et al 2018)
>
> [2] Zero-Shot Reward Specification via Grounded Natural Language (Mahmoudieh et al 2022)
>
> [3] Can Foundation Models perform Zero-Shot task Specification for Robot Manipulation (Cui et al 2022)
>
> [4] Learning Transferrable Visual Models from Natural Language Supervision (Radford et al 2021)
>
> [5] VIP: Towards Universal Visual Reward and Representation via Value-Implicit Pre-training (Ma et al 2022)
>
> [6] Language Models are Few-Shot Learners (Brown et al 2020)
>
> [7] URLB: Unsupervised Reinforcement Learning Benchmark (Laskin et al 2021)

---

> > ### Author Response · Authors · 2023-11-17
> > **Checking-in regarding further discussion**
> >
> > We hope our earlier response addressed your major concerns regarding the contribution of our work and can result in an improvement of our rating. Please inform us if there are additional questions you would like us to answer so that we can engage further before the conclusion of the discussion period.

---

> > > ### Comment · Reviewer_GnSb · 2023-11-21
> > > **Response to the authors**
> > >
> > > Dear authors,
> > >
> > > Thank you so much for your reply.
> > >
> > > After reading other reviewers' comments and your responses to us, I do have some follow up curiocity.
> > >
> > > 1. > about the pretraining design and comparison with exploration algorithms
> > >
> > > What if we allocate the optimization budget for pretraining to task-specific tuning? Will the pretrianing approach in LAMP still win against this "new baseline"? For example, if the pretraining budget is $x$, and the number of downstream tasks is $n$, then one may remove the pretraining stage but increase the budget for each task by $x/n$.
> > > This is especially a concern since the proposed method is not tested on a multitude of tasks, unlike GPT-3 [6] that you mentioned.
> > >
> > > 2. Why "the conventional goal-conditioned RL setup does not address the desiderata of scalable behavior pretraining"? For example, why couldn't one use  language prompts similar to this paper as the interested goals for behavior pretraining?
> > >
> > > 3. > "A contribution of LAMP is in identifying..."
> > >
> > > I am not sure how this "contribution" is supported in the paper. Can the authors point out related discussion/evidence in the paper?
> > >
> > > 4. I want to gently remind that several questions/concerns in my original review remain unaddressed.
> > >
> > > 5. Without any offense, I would like to say that some of the arguments/claims in the authors' rebuttal are subjective and lack of support, making the rebuttal overall less convincing. For example, the authors claim that "*The paradigm of computationally expensive, large-scale pretraining in service of multi-purpose few-shot or task-specific finetuning characterizes much of modern machine learning*." --- I am not sure how do the authors define "modern machine learning"? I think this claim only holds in foundation models and related fields.

---

> ### Author Response · Authors · 2023-11-21
> **Addressing the response and additional results**
>
> 1. Much of the allure of general purpose pretrained models is that they can be used for many downstream tasks. Our pretraining setup is designed for this in that we do not assume knowledge of the diverse downstream tasks in pretraining the agent - they are completely unseen at finetuning time. As $n$ grows, the "new baseline" proposed will become increasingly more weak as the pretraining budget per task shrinks. However, since for our method the agent is only pretrained once, the sample complexity does not scale as poorly with the number of tasks. This is because, as we demonstrate in the paper, learning each of 5 very semantically and behaviorally diverse tasks is more sample-efficient from a pretrained LAMP agent. The agent performs a closing behavior, a valve turning behavior, fine-grained picking behaviors, and a pushing behavior all benefitting from the same pretrained model. We additionally note that for many robotics tasks, seemingly small modifications to the task can harm the success rate of a pretrained agent [9]. If instead of training an agent from scratch each time, we can more efficiently learn by warm-starting training from a pretrained agent, we further realize the benefits of task-agnostic finetuning.
>
> 2. The example you describe bears relation to what we propose in LAMP and as we previously stated, our method can be seen as similar to goal-conditioned RL. However, LAMP contains novel components and examines new questions outside of the default goal-conditioned RL setting including diverse language prompting, shaped VLM rewards, pretraining task and environment design, and synergies with unsupervised objectives that differentiate the work from prior work. We are happy to acknowledge the connection of our work with goal-conditioned RL, however, maintain that LAMP involves novel contributions of interest to the community.
>
> 3. There is such discussion in the paper. In the Introduction we mention: "While these methods...they come with the novel and relatively under-explored property of being a scalable means of generating many different rewards by simply prompting with different language instructions. This property is particularly compatible with the assumptions of RL pretraining where we desire to learn general-purpose behaviors with high coverage of environment affordances, require minimal human supervision..." Also see Section 4.1, the early discussion in Section 6.1, and Section 7.
>
> 4.
>
> Regarding the 4th concern in the original response - LAMP consistently outperforms the baselines across the very different downstream tasks evaluated. The finetuning curve can be high variance, however, the trend holds averaged over seeds and across different tasks and ablations. We have added many more experiments that evidence this consistent trend in Section 6.2.
>
> Regarding the 5th concern in the original response, we have adjusted the language in the paper. We have removed the claim that "LAMP is not inherently reliant on a particular VLM" and instead clarify that LAMP can work well with CLIP rewards in addition to R3M rewards. Please see Section 6.3 of the updated paper.
>
> Regarding Question 3, we take motivation from findings in [8] that maximizing rewards from pretrained models can be a challenging optimization that suffers from reward exploitation and convergence to local minima. The paper states "In the reinforcement learning setting, an additional issue is that solely optimizing a dense reward such as VIPER can lead to early convergence to local optima." and "In the reinforcement learning literature, the entropy bonus over input trajectories corresponds to an exploration term that encourages the agent to explore and inhabit a diverse distribution of sequences within the regions of high probability under the video prediction model... VIPER is agnostic to the choice of exploration reward and in this paper we opt for Plan2Explore and RND." We similarly found empirically in our setting that adding the intrinsic novelty bonus enables the RL agent to explore more modes of a noisy reward function with many local minima and thus leads to better performance.
>
> Regarding Question 4, thank you for the catch regarding the expectation notation for the RL objective - see the revised paper with this fixed.

---

> ### Author Response · Authors · 2023-11-21
>
> For our thoughts on Question 7, see the end of the second to last paragraph of Section 6.1 on language prompting. We state "we hypothesize that both exploration coverage and exploration efficiency during pretraining play an important role. By separately increasing exploration coverage through Plan2Explore, the quality of the VLM rewards becomes more important for focusing the auxiliary exploration objective on useful exploratory behaviors. Thus, LAMP Prompt 2, which incorporates Plan2Explore, is trained on higher-quality, more relevant rewards, and can explore more efficiently during pretraining, and therefore enables more effective finetuning."
>
> 5. We simply observe the strong increase in the popularity and interest around large-scale computationally intensive pretraining for sample-efficient task-specific finetuning across machine learning (computer vision, natural language processing and generation, deep generative models, protein synthesis, robot learning, etc.) due to modern (last decade) advances in computation and datasets. We do not claim that this paradigm encompasses all of machine learning today. We situate our work within this broader trend characterized by notable breakthroughs throughout the field.
>
> We also provide some additional results to the paper that we believe strengthen the claims made in the work.
>
> We have added a new Reward Weighting Ablation section to the main text as well as added the ablation on all 5 of the evaluation tasks as opposed to just the Pick Up Cup task from the original manuscript in Section 6.2 in an effort to further enhance clarity. We arrive at a similar conclusion - LAMP is robust to this weighting and performs somewhat competitively even with an alpha value of 0 corresponding to only VLM rewards. However, the synergy of the Plan2Explore and VLM rewards leads to the optimal performance.
>
> We have similarly added a new VLM Ablation Section to the main text and evaluated on all 5 of the evaluation tasks as opposed to just Pick up Cup from the original manuscript in Section 6.3.
>
> We hope these updates have addressed your most pressing concerns and can lead to a rating increase. If not, please let us know what further clarifications or results you would require before the conclusion of the rebuttal period.
>
> [8] Video Prediction Rewards for Reinforcement Learning [Escontrela et al 2023]
>
> [9] Daydreamer: World Models for Physical Robot Learning [Wu et al 2022]

---

> > ### Comment · Reviewer_GnSb · 2023-11-22
> > **Response to the authors**
> >
> > Dear authors,
> >
> > Thank you so much for the response, which is helpful for addressing my concerns and questions. I will increase my rating to 6.

---

### Official Review · Reviewer_PM9W · 2023-10-27

**Soundness:** 3 good
**Presentation:** 3 good
**Contribution:** 2 fair
**Rating:** 5
**Confidence:** 4

**Summary:**

This paper proposes an approach for RL pre-training that relies on collecting data with VLM-based rewards functions for some semantically meaningful tasks + an exploration objective. Then, the policy is fine-tuned for the task of interest with the scripted reward and the authors show that learning is faster when pre-training is used in several environments.

**Strengths:**

The presented method is well explained and clear, the paper is well structured. The illustrations facilitate paper understanding. I find the idea of the use of VLM based rewards for semantic exploration interesting and promising. The novelty of the method is how the VLMs are used in exploration: through the noisy reward function. The ablations on the style of the prompt are interesting.

**Weaknesses:**

- I have some doubts about the usefulness of the core of the proposed method for the realistic setting. It seems that the best reported results are obtained when the semantic tasks for exploration were based on the task that was later used for fine-tuning (prompt style 2 uses reformulations of that task). How realistic is the assumption of knowing the target task at the pre-training stage? Does it mean that the method would require a new pre-training stage when the task changes? In that case, why not use the exploration budget instead for extra data collection for the task of interest? Also, the authors used the scripted reward based on the environment state for the fine tuning on the task of interest. Why not continue using the learnt reward model in this case? In a real world scenario this would be the only way to obtain the rewards.
- The paper only considers two baselines for the proposed method: training from scratch or using an exploration objective. There is a large body of literature on exploration for RL pre-training, so it would be nice to add other baselines. In particular, methods that use semantic exploration in RL with the help of pre-trained VLMs are the most relevant. See for example:
Semantic Exploration from Language Abstractions and Pretrained Representations. Allison Tam, Neil Rabinowitz, Andrew Lampinen, Nicholas A. Roy, Stephanie Chan, DJ Strouse, Jane Wang, Andrea Banino, Felix Hill. NeurIPS 2022.
- It seems that the proposed method includes the component of semantic exploration with only 0.1 weight and the rest is based on the prior work Plan2Explore. I am wondering how sensitive the method is to the choice of this weight? If the proposed objective is used without Plan2Explore, does it still outperform the two baselines?

Some minor points:
- Why was it necessary to do scene augmentations with the same textures/images that were used for pre-training a VLM (Ego4D)? Could other (independent of pre-training) scene augmentations be used?
- The robotic scene is not taken into account when semantic tasks at the pre-training stage are selected and as authors notes, some of them are infeasible or even meaningless (prompt style 6). Then, would it be sufficient in that case to replace the rewards for exploration with just random rewards?

**Questions:**

I would like to hear the authors' discussion regarding the three weaknesses that I highlighted above:
- usefulness of the method given finetuning on similar tasks and use of scripted rewards for finetuning;
- baselines for explorations with the help of VLMs;
- the effect of constant 0.1 and the importance of prior explorations objectives.

---

> ### Author Response · Authors · 2023-11-12
> **Key Clarification of Method and Resulting Contribution, Desired Result Already Present in Supplementary Material**
>
> **Important Clarification of Pretraining Setting and Contribution**
>
> It appears that Reviewer PM9W may have missed that the pretraining phase is fully finetuning task-agnostic and a “one-time cost.” During the pretraining phase, the agent optimizes different rewards and encounters many different but randomly-sampled tasks. The analysis we perform during pretraining examines various prompting styles and their relevance to the pretraining tasks, which we assume knowledge of for certain prompting styles, and not the finetuning tasks, which we assume no knowledge of. Prompt Style 2 is not based on the finetuning tasks but the pretraining tasks. VLM-guided RL pretraining is only done once, but the same, single pretrained agent is able to achieve the improved finetuning results on all the tasks evaluated. The 5 finetuning evaluation tasks are not included in the pretraining phase. This setup was intentional as this line of work is in pursuit of a single, general-purpose pretrained RL agent that can adapt to arbitrary new tasks with better sample-efficiency. We expect this important clarification to change your evaluation of the contribution as well address your questions. Please inform us if you have further questions.
>
> Using LAMP directly as a task reward would not enable the RL agents to solve any of the tasks. As we evidence in Figure 3, and motivate in the introduction, rewards from current pretrained VLMs are far too noisy and susceptible to reward exploitation to be usable with online RL. In fact, this is central to our message - current VLMs are ill-suited for this purpose in that they lack the required precision to supervise challenging robot manipulation tasks, however, they do possess certain properties that render them very useful for pretraining RL agents with limited supervision. Scripted finetuning task rewards are simply a means to provide a consistent evaluation of the adaptation capabilities of the various pretrained agents. Our contribution is centered on proposing an RL agent pretraining setup.
>
> We thank the reviewer for mentioning an additional baseline. While this baseline is relevant and worth citation, it makes additional assumptions on the environment - in this case access to an oracle language captioner which provides perfect natural langue descriptions of the current visual scene. We direct the reviewer to an additional widely-used baseline with more similar environment assumptions to our own in Random Network Distillation (RND), which is also baselined against in the mentioned work. We provide results in the Appendix (Figure 13, Section D.3.1). LAMP is able to outperform this baseline on all of the tasks evaluated.
>
> [1] Exploration by Random Network Distillation (Burda et al 2018)
>
> **Desired Results in Appendix**
>
> We were similarly interested to understand the impact of the reward weighting term. We had already provided these results in the Appendix. Please refer to Figure 11 in Section D.2 of the Supplementary Material. We find that LAMP works best with our choice of alpha but still exhibits very strong performance for an alpha value of 0 corresponding to training only with VLM rewards and no Plan2Explore bonus. We think this is quite an encouraging result and hope this addresses the reviewers concerns.
>
> **Ego4D Texture Overlay**
>
> We used the Ego4D textures during pretraining to increase the difficulty of the control task with more complex visuals to more closely approximate the real world. These textures help to narrow the domain gap between the robot environment and the datasets used to pretrain the VLMs. We adopt the same texture randomization scheme for all pretraining methods we evaluate for fair comparison.

---

> > ### Author Response · Authors · 2023-11-17
> > **Checking-in regarding further discussion**
> >
> > We hope our earlier response addressed your major concerns regarding the contribution of our work and can result in an improvement of our rating. Please inform us if there are additional questions you would like us to answer so that we can engage further before the conclusion of the discussion period.

---

> > > ### Author Response · Authors · 2023-11-22
> > > **Additional Results**
> > >
> > > We have added additional experiments we thought to make the reviewer aware of.
> > >
> > > We have added a new Reward Weighting Ablation section to the main text as well as added the ablation on all 5 of the evaluation tasks as opposed to just the Pick Up Cup task from the original manuscript in Section 6.2 in an effort to further enhance clarity. We arrive at a similar conclusion - LAMP is robust to this weighting and performs somewhat competitively even with an alpha value of 0 corresponding to only VLM rewards. However, the synergy of the Plan2Explore and VLM rewards leads to the optimal performance.
> > >
> > > We have similarly added a new VLM Ablation Section to the main text and evaluated on all 5 of the evaluation tasks as opposed to just Pick up Cup from the original manuscript in Section 6.3.
> > >
> > > We look forward to hearing about any remaining questions or concerns before the conclusion of the rebuttal period.

---

### Official Review · Reviewer_hH26 · 2023-10-29

**Soundness:** 3 good
**Presentation:** 3 good
**Contribution:** 3 good
**Rating:** 6
**Confidence:** 3

**Summary:**

This paper proposes LAMP, a pre-training algorithm that uses the reward from R3M plus plan2explore rewards to pre-train the agents, and then finetunes the agents with task rewards. LAMP achieves better sample efficiency on 5 tasks from RLBench, and ablation results show that the correct language prompt and the vision-language model are key to the success.

**Strengths:**

The usage of R3M is overall interesting. Since the reward from out-of-domain models is noisy naturally (Figure 3), this work proposes to use such noisy signals as a pre-training method. Also, some factors presented to enhance such rewards, including plan2explore rewards and language prompts, are natural and well-motivated.

Overall I think this paper is interesting and would tend to give acceptance.

**Weaknesses:**

- **The weight of plan2explore rewards.** The authors use a weight of 0.9 for plan2explore rewards, and only a weight of 0.1 for their proposed rewards. Would the weight impact much? It could be interesting to see more ablation results for this parameter.
- **Lack of baseline**. I think VIPER [1] and VIP [2] could also serve as baselines. It would be good to see more baseline results in Figure 5.
- **No results about pre-training stages.** What is the success rate of directly applying rewards of LAMP for in-domain task learning?

[1] Escontrela, Alejandro, et al. "Video Prediction Models as Rewards for Reinforcement Learning." arXiv preprint arXiv:2305.14343 (2023).


[2] Ma, Yecheng Jason, et al. "Vip: Towards universal visual reward and representation via value-implicit pre-training." arXiv preprint arXiv:2210.00030 (2022).

**Questions:**

See **weakness**.

---

> ### Author Response · Authors · 2023-11-11
> **Major Desired Result Already Present in Supplementary Material**
>
> **Desired Results in Appendix**
>
> We were similarly interested to understand the impact of the reward weighting term. We had already provided these results in the Appendix. Please refer to Figure 11 in Section D.2 of the Supplementary Material. We find that LAMP works best with our choice of alpha but still exhibits very strong performance for an alpha value of 0 corresponding to training only with VLM rewards and no Plan2Explore bonus. We think this is quite an encouraging result and hope this addresses the reviewers concerns.
>
> **Other Pretrained Models**
>
> VIPER is not a VLM in that it is not language-conditioned. Also our focus is on leveraging pretrained VLMs with strong generalization capabilities by providing diverse prompting to generate diverse rewards. VIPER is only shown to work as a reward function on mostly in-domain data. VIP, although it demonstrates some level of domain generalization, is also not a VLM, but rather relies on goal image alignment to compute a reward. Using such a method in a pretraining scheme quite different from that to which we propose would possibly lead to interesting results, however, would require possessing many interesting and diverse exploration goals a priori, which is a strong assumption of open-ended pretraining. Central to our contribution is that diverse language is a uniquely flexible means of generating diverse rewards with relatively little human supervision. In general, the message of our work is that providing diverse prompting with pretrained VLMs to generate rewards can serve as an effective pretraining scheme, not that R3M is the only particular pretrained model that can work.
>
> **LAMP as a task-reward**
>
> To answer simply, the success rate when using LAMP as a task reward on all of the tasks is 0%. As we evidence in Figure 3, and motivate in the introduction, rewards from current pretrained VLMs are far too noisy and susceptible to reward exploitation to be usable with online RL. In fact, this is central to our message - current VLMs are ill-suited for this purpose in that they lack the required precision to supervise challenging robot manipulation tasks, however, they do possess certain properties that render them very useful for pretraining RL agents with limited supervision.

---

> > ### Author Response · Authors · 2023-11-17
> > **Checking-in regarding further discussion**
> >
> > We hope our response addressed your major concerns and can result in an improvement of our rating. Please inform us if there are additional questions you would like us to answer so that we can engage further before the conclusion of the discussion period.

---

> ### Comment · Reviewer_hH26 · 2023-11-21
> **Thank the authors for the response.**
>
> Thank the authors for the response.
>
> ## Other Baselines
> It is obvious that the baselines mentioned differ from the proposed method (e.g., not VLM), but the similarity is that they could all provide reward signals based on observations. Though these reward functions are designed for in-domain tasks, they are also noisy compared to the oracle reward and thus could also serve as a pertaining method.
>
> Therefore, it is still good to see more comparison with these recent works.
>
> ## LAMP as a task-reward
> > To answer simply, the success rate when using LAMP as a task reward on all of the tasks is 0%.
>
> It is good to see the points described by the authors are updated in the main paper.
>
> -----
>
> Again, thank the authors for their response. I do not have further questions.

---

> > ### Author Response · Authors · 2023-11-21
> > **Thanks for Acknowledging and further experiments**
> >
> > We thank the reviewer for acknowledging the additional clarity provided by our rebuttal.
> >
> > We also provide some additional results to the paper that we believe strengthen the claims made in the work.
> >
> > We have added a new Reward Weighting Ablation section to the main text as well as added the ablation on all 5 of the evaluation tasks as opposed to just the Pick Up Cup task from the original manuscript in Section 6.2 in an effort to further enhance clarity. We arrive at a similar conclusion - LAMP is robust to this weighting and performs somewhat competitively even with an alpha value of 0 corresponding to only VLM rewards. However, the synergy of the Plan2Explore and VLM rewards leads to the optimal performance.
> >
> > We have similarly added a new VLM Ablation Section to the main text and evaluated on all 5 of the evaluation tasks as opposed to just Pick up Cup from the original manuscript in Section 6.3.
> >
> > We hope these updates have addressed your most pressing concerns and can lead to a rating increase. If not, please let us know what further clarifications or results you would require before the conclusion of the rebuttal period.

---

### Official Review · Reviewer_YXuM · 2023-10-31

**Soundness:** 2 fair
**Presentation:** 3 good
**Contribution:** 2 fair
**Rating:** 5
**Confidence:** 3

**Summary:**

The paper proposes a method to pretrain a language-conditioned RL policy using feedback from a vision-language model (that aligns video frames with the task completion) combined with an exploration bonus. Such a policy is then finetuned on individual RL tasks. Performance is analyzed across a few different tasks, and components of the method are ablated.

**Strengths:**

- The proposed method shows a solid benefit on some of the evaluation tasks, and at least a minor benefit on all (Figure 5)
- The writing is in general quite clear and easy to follow

**Weaknesses:**

- As far as I can tell, the main contribution here is the choice of vision-language model, the particular temporal objective (a minor change), and the addition of an exploration reward (Plan2Explore), but if the authors claim that this method is worth further study regardless of the specifics in the paper, they need to be ablated. The exploration reward is ablated (RND shows up as standalone pretraining in Figure 13). The vision-language model is only ablated on a single task (Figure 7). The objective is never ablated, unless we count the comparison to ZeST as a combination, in which case it's unclear that there is a benefit to the proposed scheme.
- Regardless of claims of novelty, the result is not super strong. From Appendix C.4 and Figure 12 it is my understanding that pretraining is a minimum of 100k steps, which would be nice to see in the main paper. From Figure 5, we see that in the worst case, pretraining saves us 50k steps (Pick Up Cup, Take Lid Off Saucepan) and pretraining with P2E is quite close. Combine this with the missing baselines and I wonder if the method is competitive on a fixed-compute budget with single-task training.
- I'm not sure if the evaluations are separated into in-distribution and out-of-distribution tasks, but it would be worthwhile to understand the performance drop on OOD tasks that were not used in pretraining to further understand the benefit of this pretraining.
- I think it would be worthwhile to include a baseline that is finetuned from some supervised language-conditioned model to further justify why we need to pretrain in the RL setting. RL is complex and unwieldy, so it would be nice to justify its necessity through an experiment.
- There are many moving parts in this method (proposed method, Masked World Model policy, Plan2Explore additional reward, architectural choices against baselines, notice the number of hyperparameters in Tables 5, 6, 7, 8 and 9) which are significantly more complex than the standard deep RL setting, and it would be nice to see experiments justifying all of these choices, or at least further argument.

It is possible I missed some of the answers to these points on the first read, so please do point them out if so.

**Questions:**

- How many tasks are used for pretraining and evaluation?
- Is there a split in the evaluation into seen and unseen tasks? This was the original motivation for the method so I think it is worthwhile to test.

---

> ### Author Response · Authors · 2023-11-11
> **Clarification that Pretraining is Task-Agnostic and "one-time cost"**
>
> **Important Clarification of Pretraining Setting and Contribution**
>
> It appears that Reviewer YXuM may have missed that the pretraining phase is fully finetuning task-agnostic and a “one-time cost.” During the pretraining phase, the agent optimizes different rewards and encounters many different but randomly-sampled tasks. The analysis we perform during pretraining examines various prompting styles and their relevance to the pretraining tasks, which we assume knowledge of for certain prompting styles, and not the finetuning tasks, which we assume no knowledge of. The above point addresses “pretraining saves us 50k steps…” and “I wonder if the method is competitive on a fixed-compute budget with single-task training.” VLM-guided RL pretraining is only done once, but the same, single pretrained agent is able to achieve the improved finetuning results on all the tasks evaluated. The 5 finetuning evaluation tasks are not included in the pretraining phase. This setup was intentional as this line of work is in pursuit of a general-purpose pretrained RL agent that can adapt to arbitrary new tasks with better sample-efficiency. Furthermore, to the third point, all 5 of the evaluations are on OOD tasks. The pretraining tasks are randomly sampled from a set of combinatorially many possible tasks using ShapeNet [2] objects. See Figure 8 in Appendix C.2 for examples. We expect this important clarification to change your evaluation of the contribution as well address your questions. Please inform us if we have misinterpreted your concerns or if you have further questions.
>
> **Ablations**
>
> Regarding the ablations, Reviewer YXuM mentions that we do ablate the choice of exploration reward with inclusion of RND in the Appendix. We agree with this and are glad the reviewer has acknowledged this ablation. Furthermore, they suggest ablating the choice of choice of temporal objective in order to prove that there is “benefit to the proposed scheme.” We do make strong claims that a particular formulation of temporal alignment is uniquely critical for pretraining, rather we propose in the paper that it be used as a language-conditioned pretraining reward and demonstrate the benefits when integrating it in LAMP. We simply employ the most natural alignment objective based on the way the VLM is trained or what has been found to work well in prior work. If the reviewer still disagrees we would appreciate further explanation and suggestions as to what other language-temporal alignment objectives they would like to see.
>
> To address the last point, the hyperparameter choices for the Masked World Models agent follow the paper which introduced the method [1]. We treat the RL largely as a black box and do not tune RL specific agent hyperparameters as these are orthogonal to our method. We provide the default hyperparameters in the interest of enhancing simple reproducibiilty. The base RL algorithm is kept entirely constant across all baselines (Plan2Explore, RND). The justification for using Masked World Models (MWM) as opposed to a “more standard” RL algorithm such as SAC, PPO, DreamerV3 is that MWM can solve all of the challenging downstream tasks evaluated from scratch with no tuning. We found that this was not the case for the aforementioned more standard algorithms. Lastly, we point the reviewer to our ablation on the balance weighting between exploration reward and VLM reward in Appendix Section D.2 Figure 11 as this hyperparameter is related to our contribution.
>
> [1] Masked World Models for Visual Control (Seo et al 2022)
>
> [2] ShapeNet: An Information Rich 3-D Model Repository (Chang et al 2015)

---

> > ### Author Response · Authors · 2023-11-17
> > **Checking-in regarding further discussion**
> >
> > We hope our earlier response addressed your major concerns regarding the contribution of our work and can result in an improvement of our rating. Please inform us if there are additional questions you would like us to answer so that we can engage further before the conclusion of the discussion period.

---

> > > ### Comment · Reviewer_YXuM · 2023-11-20
> > > **Thanks for the clarifications**
> > >
> > > > It appears that Reviewer YXuM may have missed...
> > >
> > > I did not miss this point. I do not think the current gains in sample efficiency in the paper are significant enough to merit the extra complexity. Typically pretraining on WikiText-103 can yield efficiency gains of orders of magnitude in language tasks, yet here we have <50% savings on a few tasks. Especially as the pretraining is relatively cheap compared to the finetuning, it may not make sense to even do pretraining in these tasks, or perhaps a supervised pretraining method would be more cost-efficient than requiring the existence of a randomized simulator.
> > >
> > > > The 5 finetuning tasks are not included in the pretraining phase...
> > >
> > > Can the authors point me to where this is clarified in the text? If not, then it definitely should be in there. In Section 5.1, where I might expect such a clarification, it doesn't appear. Also it's not clear whether the tasks are "unseen" or "OOD," which the response indicates they are somewhat unlikely in the pretraining distribution, but not explicitly disallowed.
> > >
> > > > ...ablate the choice of exploration reward...
> > >
> > > This should be referenced in the main text. Having to dig for it is undesirable in order to understand the contributions of the work.
> > >
> > > > ...suggestions as to what other language-temporal alignment objectives they would like to see.
> > >
> > > R3M is not the only VLM. There are even ZeST experiments in the Appendix. These kinds of ablations should appear in the main text. I would also like to see a comparison, for example against Voltron (https://arxiv.org/pdf/2302.12766.pdf), or justification for the choice of RL vs. non-RL pretraining
> > >
> > > > To address the last point...
> > >
> > > Thanks for these clarifications, I agree that it is fine to consider these as a black box, but it was not clear from the text why such methods were chosen.
> > >
> > > > ...our ablation on the balance weighting...
> > >
> > > Thanks for the pointer. I agree that this clarifies somewhat, yet it is (to my knowledge) not referenced in the main text. So it is unclear how I would arrive at these conclusions without the extra help.

---

> ### Author Response · Authors · 2023-11-21
> **Thanks for Acknowledging Clarification. Additional Results**
>
> **Setting and Contribution**
>
> We are glad that the reviewer understands that the pretraining is a one-time cost aimed at pretraining a general purpose agent. In this work, we simply build on a wide-body of literature aimed at improving general purpose RL pretraining with limited human supervision. We mention many of these kinds of works in the Pretraining for RL section of the Related Works and situate our contribution therein. The reviewer mentions orders of magnitude gains in efficiency for language tasks by referencing Internet-scale language pretraining. They go on to reason that our much more modest gains do not warrant the additional complexity of our method. However, it is safe to say that the domains of reinforcement learning and robotics are far behind language in terms of the adaptive capabilities of general-purpose models. Recent work such as [3] makes this very apparent. A longstanding challenge in RL and robotics is to build similarly capable models, though current methods fall well short. We far from claim to rival these levels of capability, rather we propose an approach that improves on current methods and reveals potential avenues for further improvements.
>
> **Improved Clarity and New Experimental Ablations**
>
> We have clarified our experimental setting by adding "We do not include any of the downstream evaluation tasks that the agent finetunes on in the pretraining setup. The pretraining tasks are exclusively comprised of ShapeNet objects. All of the downstream tasks are unseen tasks from the original RLBench task suite." Please see the updated Section 5.1.
>
> Regarding the use of MWM, in Section 4.2, we justify our decision with "we use Masked World Models (MWM) (Seo et al.,
> 2022), an off-policy, model-based method with architectural inductive biases suitable for fine-grained robot manipulation."
>
> Thank you for confirming the additional clarity provided by the reward weighting ablation. We have added a new Reward Weighting Ablation section to the main text as well as added the ablation on all 5 of the evaluation tasks as opposed to just the Pick Up Cup task from the original manuscript in Section 6.2 in an effort to further enhance clarity. We arrive at a similar conclusion - LAMP is robust to this weighting and performs somewhat competitively even with an alpha value of 0 corresponding to only VLM rewards. However, the synergy of the Plan2Explore and VLM rewards leads to the optimal performance.
>
> We have similarly added a new VLM Ablation Section to the main text and evaluated on all 5 of the evaluation tasks as opposed to just Pick up Cup from the original manuscript in Section 6.3.
>
> We hope these updates have addressed your most pressing concerns and can lead to a rating increase. If not, please let us know what further clarifications or results you would require before the conclusion of the rebuttal period.
>
> [3] GPT-4 Technical Report [OpenAI 2023]

---

> > ### Comment · Reviewer_YXuM · 2023-11-22
> >
> > Thanks for the additional clarifications. I'll respond to the lingering difficulties below:
> >
> > > ...referencing Internet-scale language pretraining...
> >
> > It is certainly unfair to compare small-scale pretraining to internet-scale. I would think internet-scale might mean something like LAION (5b images) or RedPajama-v2 (30t text tokens). Wikitext-103 is not internet-scale, rather only 103m text tokens and very workable in single-GPU or small cluster settings.
> >
> > I agree that reinforcement learning is behind, but I don't find the results that convincing. I would even prefer a longer pretraining period if it were demonstrated that such process leads to even stronger results. With a sufficiently small constant factor (100k steps), the extra computation might not matter that much vs. the added complexity of pretraining. Especially as the pretraining requires a unified simulator/robot setup that might not be available for every new task one wants to complete.
> >
> > > ...added a new Reward Weighting Ablation...
> >
> > I appreciate the additional experiment, but it seems preliminary for a few reasons. First, the full range of alpha values is only presented for 1 domain, and second the caption is lacking in detail.
> >
> > > ...added a new VLM Ablation Section...
> >
> > This is a step in the right direction. I think the caption should have a bit more precise discussion, but if it overlaps exactly with the text perhaps that's a waste of space.
> >
> > I should add that currently the text is now over the 9 page limit, so it is somewhat unfair to review in this format compared to the required 9 pages. I think one option would be to shrink Figure 2 a bit, condense Figures 5, 6 and 8 to single rows, perhaps with a shared legend.

---

> > > ### Author Response · Authors · 2023-11-23
> > > **Resolving lingering difficulties and further rebuttal**
> > >
> > > Thanks for responding with your lingering difficulties. We provide an updated paper and supplementary material to address them as well as further rebuttal below:
> > >
> > > **Internet-scale language pretraining**
> > >
> > > Regarding large-scale pretraining in natural language and the reviewer's comment that "Typically pretraining on WikiText-103 can yield efficiency gains of orders of magnitude in language tasks":
> > >
> > > The reviewer in essence argues that since there can be large performance boosts from an 103m token language dataset "in language tasks," that our smaller performance boosts in visuomotor control via RL pretraining are unconvincing. We find this notion somewhat problematic for a few reasons. Primarily, the surprisingly high complexity associated with skill learning in the control domain compared to the perceptual and abstract reasoning domains (vision, language) has been widely observed in the field (Moravec's Paradox) [5]. For instance, state-of-the-art skill discovery methods, which stand on long-standing research in option-learning and unsupervised RL, are still only able to discover rather primitive locomotion and more recently manipulation behaviors from low-level simulator state after tens of millions of samples [3][4]. This is no comparison to the language or image domains where the field has developed effective unsupervised learning methods for learning rich representations from datasets - presumably as you have referenced regarding WikiText-103.
> > >
> > > Furthermore, the reasoning challenges present in the control domain are in many ways different and thus can be more data intensive. Closed-loop control in the RL setting involves credit assignment, exploration in complex environments with long-horizons, and non-stationarity on top of the more traditional supervised learning components common in NLP and vision. Ultimately, directly equating the necessary pretraining scales in the language domain to the control domain is an unfair means of evaluating contributions in RL where we are yet to see such orders of magnitude gains in data-efficiency.
> > >
> > > Lastly, the general reference to "pretraining on WikiText-103" is still very vague. We would appreciate if the reviewer could at all describe the experiment they are referencing and/or provide a reference to the work that proposes it. It is quite difficult to further contextualize or defend our contribution with the current level of information provided by the reviewer.
> > >
> > > **Pretraining Complexity**
> > >
> > > The reviewer states "the pretraining requires a unified simulator/robot setup that might not be available for every new task one wants to complete" in critiquing the "added complexity of pretraining" associated with our contribution. We re-emphasize that our pretraining experimental setup is by design detached from the very diverse finetuning tasks we select. We demonstrate gains in downstream sample-efficiency on tasks that are outside of the pretraining distribution both perceptually and in terms of the kinds of behaviors required. Therefore, we in fact do not require a "unified simulator/robot setup" for "every new task we want to complete."
> > >
> > > We amend the supplementary material to include that we used 300k steps of pretraining for our main experiments and the added ablations in the main paper.
> > >
> > > **Presentation: page-limit, captions, ablations**
> > >
> > > In response to the critique that we sweep the alpha hyperparameter on a single task, we point out that perhaps the most informative reward-weighting ablation is that which we include on all 5 tasks - the relative contribution of the VLM weight vs. the Plan2Explore novelty bonus. We demonstrate on all tasks that the VLM weight in isolation produces competitive results to the Plan2Explore setup. This clarifies that our contribution is not marginal, rather actually contributes to finetuning on diverse tasks without the help of prior pretraining methods. Furthermore, we show that by selecting a single hyperparameter selection of alpha=0.9, we can obtain a synergy of the two reward signals across all tasks.
> > >
> > > As the reviewer suggested, we add more precise captions to the new ablations as well as reduce the paper to be under the 9-page limit by resizing figures for fair evaluation.
> > >
> > > We hope we have satisfied the reviewer's lingering questions regarding our work to warrant an increase to our original rating.
> > >
> > > [3] Controllability-Aware Unsupervised Skill Discovery [Park et al 2023]
> > >
> > > [4] Lipschitz-constrained Unsupervised Skill Discovery [Park et al 2022]
> > >
> > > [5] Mind Children [Moravec 1988]

---

### Meta-Review · Area_Chair_akdH · 2023-12-06

**Metareview:**

The paper proposes LAMP, a method to pretrain a language-conditioned RL policy using feedback from a vision-language model (that aligns video frames with the task completion) combined with an exploration bonus from plan2explore. The resulting policy is then finetuned on individual RL tasks downstream. LAMP achieves better sample efficiency on 5 tasks from RLBench, and ablation results show that the correct language prompt and the vision-language model play a significant role it its success. The reviewers found that the paper did not sufficiently clarify the central contribution, and this was somewhat unclear even after the discussion period. The paper would also benefit from providing a detailed description of the pre-training stage, and additional comparisons with baselines,

**Justification For Why Not Higher Score:**

Reviewers found it difficult to identify the primary contribution in the paper as presented. There are also insufficient comparisons with baselines, and a lack of information about the pre-training stage.

**Justification For Why Not Lower Score:**

N/A

---

### Decision · Program_Chairs · 2024-01-16

Reject